# Mercury evidence from southern Pangea terrestrial sections for end-Permian global volcanic effects

Jun Shen [1] ✉, Jiubin Chen [2], Jianxin Yu [3], Thomas J. Algeo[1,3,4], Roger M. H. Smith[5,6], Jennifer Botha[5,7], Tracy D. Frank [8], Christopher R. Fielding[8], Peter D. Ward[9] & Tamsin A. Mather [10]

The latest Permian mass extinction (LPME) was triggered by magmatism of the Siberian Traps Large Igneous Province (STLIP), which left an extensive record of sedimentary Hg anomalies at Northern Hemisphere and tropical sites. Here, we present Hg records from terrestrial sites in southern Pangea, nearly anti-podal to contemporaneous STLIP activity, providing insights into the global distribution of volcanogenic Hg during this event and its environmental processing. These profiles (two from Karoo Basin, South Africa; two from Sydney Basin, Australia) exhibit significant Hg enrichments within the uppermost Permian extinction interval as well as positive $\Delta^{199}Hg$ excursions (to ~0.3‰), providing evidence of long-distance atmospheric transfer of volcanogenic Hg. These results demonstrate the far-reaching effects of the Siberian Traps as well as refine stratigraphic placement of the LPME interval in the Karoo Basin at a temporal resolution of ~$10^5$ years based on global isochronism of volcanogenic Hg anomalies.

The Siberian Traps Large Igneous Province (STLIP) is regarded as the ultimate trigger for the latest Permian mass extinction (LPME, ca. 252 Ma) and associated global-scale environmental perturbations. High-resolution zircon U–Pb dating records make the Permian–Triassic transition one of the most well-studied intervals for cause-and-effect-relationships between LIP volcanism and ecosystem perturbations[1–3]. However, U–Pb dating relies on the use of zircons in volcanic ash beds. The widespread occurrence of such zircons is mainly in the Paleo-Tethys region[4,5], limiting the linkage of environmental and biological signals to well-dated fossiliferous marine successions. Although geochemical proxies such as the concentrations and isotopes of Zn[6] and Ni[7–9] have been employed to track volcanic inputs to Permian–Triassic transitional successions more widely, their utility in this regard is limited by facies dependency and susceptibility to surficial recycling processes.

For the past decade, mercury (Hg) enrichments and isotopes have been widely used to track volcanic inputs to both marine and terrestrial environments during the Permian–Triassic transition (Fig. 1 and Supplementary Fig. 1) as well as other intervals[10–12]. Massive volcanogenic Hg inputs can overwhelm normal buffering mechanisms, leading to spikes in both raw and normalized Hg concentrations (i.e., ratios of Hg to total organic carbon, Hg/TOC) in sediments[10–13]. Sediment Hg enrichments associated with the LPME at far-flung sites (e.g., $n = 38$, Fig. 1, Supplementary Fig. 1) document a synchronous widespread increase in Hg fluxes[13,14]. Furthermore, Hg isotopes, especially $\Delta^{199}Hg$, which is stable (exhibiting no or limited fractionation) under

[1]State Key Laboratory of Geological Processes and Mineral Resources, China University of Geosciences, Wuhan, Hubei 430074, People's Republic of China. [2]Institute of Surface-Earth System Science, Tianjin University, Tianjin 300072, People's Republic of China. [3]State Key Laboratory of Biogeology and Environmental Geology, China University of Geosciences, Wuhan, Hubei 430074, People's Republic of China. [4]Department of Geosciences, University of Cincinnati, Cincinnati, OH 45221-0013, USA. [5]Evolutionary Studies Institute, University of the Witwatersrand, Johannesburg 2050, South Africa. [6]Iziko South African Museum, PO Box 61 Cape Town 8000, South Africa. [7]National Museum, PO Box 266 Bloemfontein 9300, South Africa. [8]Department of Earth Sciences, University of Connecticut, Storrs, CT 06269, USA. [9]Department of Biology, University of Washington, Seattle, WA 98195-1800, USA. [10]Department of Earth Sciences, University of Oxford, South Parks Road, Oxford OX1 3AN, UK. ✉e-mail: shenjun@cug.edu.cn

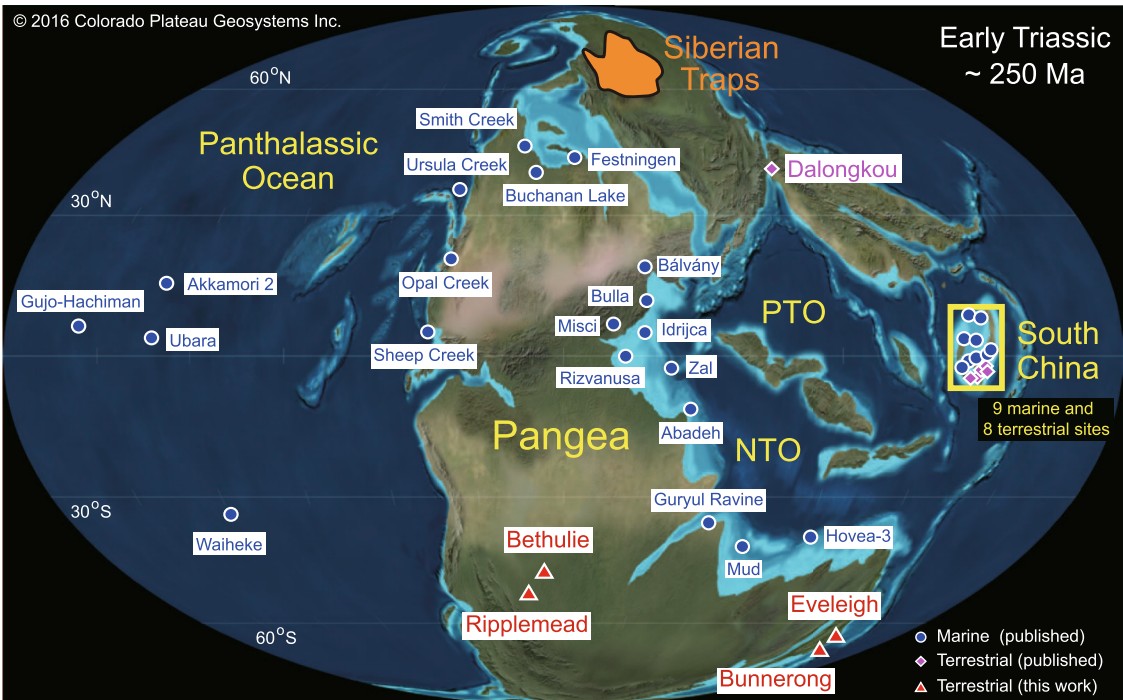

**Fig. 1 | Global paleogeography of the Early Triassic (~250 Ma).** Adapted from Ron Blakey, http://jan.ucc.nau.edu/-rcb7/, © 2016 Colorado Plateau Geosystems Inc. Red triangles represent the study sites. The Bethulie and Ripplemead sections are located in the mid-latitude Karoo Basin, South Africa (paleo -30–60°S), and the Eveleigh and Bunnerong sections are located in the high-latitude Sydney Basin, Australia (paleo -60°–90°S). Circles and diamonds represent other marine and continental sections, respectively, for which mercury data have been generated. The name and locations of 9 marine and 8 terrestrial Permian–Triassic boundary sites in South China are shown in Supplementary Fig. 1, and the sources for each site are given in Supplementary Table 1.

diverse physical, chemical, and biological conditions, can provide information regarding Hg provenance and processing[15]. Earlier studies have demonstrated that volcanogenic Hg sources tend to dominate in offshore settings[14,16] and terrestrial Hg sources in nearshore and terrestrial facies[16–19]. However, terrestrial sections have received far less attention ($n = 9$) compared to marine sections ($n = 29$; Fig. 1, Supplementary Fig. 1). In addition, most published records are from the paleo-Northern Hemisphere and Equatorial regions (especially South China, which accounts for 8 out of 9 terrestrial records) with no studies from the paleo-Southern Hemisphere (Fig. 1 and Supplementary Fig. 1). This geographically uneven distribution of Hg records limits our ability to assess the global footprint and impacts of the STLIP eruptions.

In this study, we present Hg concentration data from four terrestrial sites in the Southern Hemisphere, including two cores (Bunnerong, 33.97°S, 151.23°E and Eveleigh, 33.90°S, 151.19°E) from the Sydney Basin, Australia, and two outcrop sections (Ripplemead, 31.49°S, 24.25°E and Bethel, 30.42°S, 26.26°E) from the Karoo Basin, South Africa (Fig. 1). To date, these are the most distant terrestrial sites from the STLIP to have been investigated for volcanogenic Hg, which was located at high northern paleolatitudes (50–80°N). We also analyzed Hg isotopes for a subset of samples from the Bunnerong and Ripplemead sections in order to track Hg sources. Furthermore, our work provides constraints on the stratigraphic position of the LPME interval in the Karoo Basin, which has been a subject of uncertainty (n.b., in thick terrestrial successions, this biological event spans a discrete stratigraphic interval rather than being recorded at a single horizon).

## Results
### Mercury concentration records
In the Sydney Basin, both carbon isotopic compositions of organic matter ($\delta^{13}C_{org}$) and ratios of mercury to total organic carbon (Hg/TOC) exhibit significant variations through the Permian–Triassic

transition (Fig. 2 and Supplementary Fig. 2). In the Bunnerong core (Fig. 2a), the Upper Permian exhibits relatively uniform $\delta^{13}C_{org}$ values of -–24‰ and Hg/TOC of <30 ppb/%, followed by rapid shifts near the Permian–Triassic boundary to −29‰ and >100 ppb/%, respectively. Almost identical patterns are observed in the Eveleigh core (Fig. 2b), with a few Hg/TOC peaks (to -100 ppb/%) in the Lower Triassic. In the Karoo Basin, $\delta^{13}C_{org}$ at Ripplemead varies from −27 ‰ to −24 ‰ with more negative values spanning the Permian–Triassic transition, accompanying an increase in Hg/TOC (to -50-100 ppb/%) from background values of <40 ppb/% (Fig. 2c, Supplementary Fig. 3). At Bethulie, background Hg/TOC is <100 ppb/% but peaks to -1000 ppb/% are observed during the Permian–Triassic transition (Fig. 2d and Supplementary Fig. 3).

### Mercury isotope records
The mass independent fractionation (MIF) of odd Hg isotopes ($\Delta^{199}Hg$) exhibits a similar increasing trend through the LPME in both study areas (Fig. 3). At Bunnerong (Sydney Basin), $\Delta^{199}Hg$ shows a positive shift from the pre-LPME Upper Permian (+0.1 to +0.2‰) to the Permian–Triassic transition interval (-–0.1‰) (Fig. 3a), whereas at Ripplemead (Karoo Basin), $\Delta^{199}Hg$ values are close to 0‰ or slightly negative both below and above the LPME, with a significant increase to +0.1 to +0.2‰ within the Permian–Triassic transition interval (Fig. 3b).

## Discussion
Hg concentration data must be normalized to some other sediment parameter to avoid influences related to lithologic variation, in order to meaningfully assess levels of Hg enrichment[12]. Recognition of the dominant host phase of Hg is essential for normalization of Hg concentrations. Because organic matter is the most common host phase of Hg, normalization to total organic carbon (Hg/TOC) has been widely used[11], but other phases such as sulfides (proxied by total sulfur, TS) or clay minerals (proxied by aluminium, Al) can host Hg under some

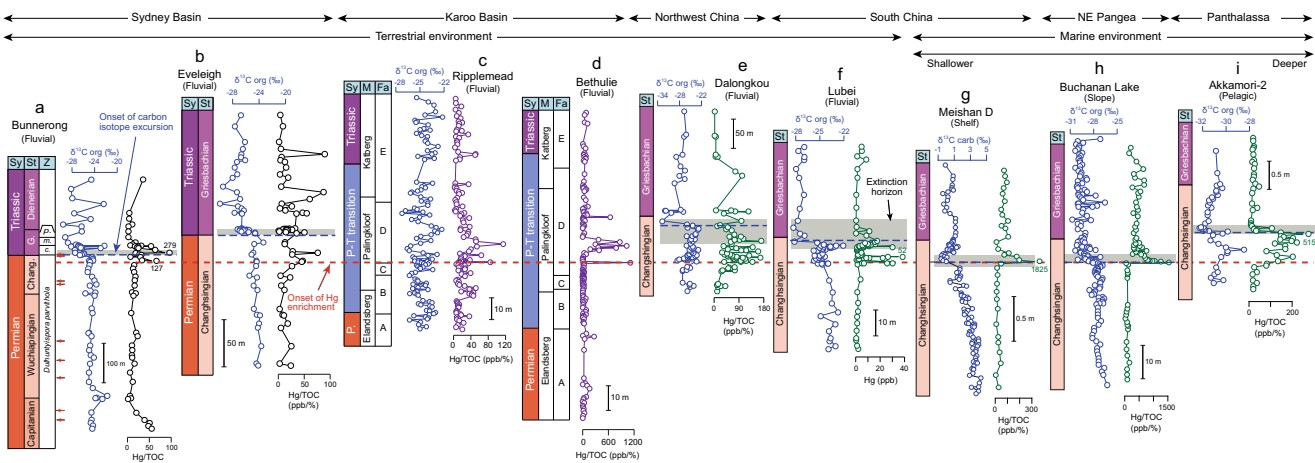

**Fig. 2 | Timeframe, carbon isotope compositions of organic matter ($\delta^{13}C_{org}$, ‰) and carbonate ($\delta^{13}C_{carb}$, ‰), and mercury to total organic carbon (Hg/TOC, ppb/%) profiles of selected global sections. a** Bunnerong (Fielding et al.[35] and this study for $\delta^{13}C_{org}$ and Hg/TOC, respectively); **b** Eveleigh (Fielding et al.[57] and this study for $\delta^{13}C_{org}$ and Hg/TOC, respectively); **c** Ripplemead (this study); **d** Bethulie (this study); **e** Dalongkou[17]; **f** Lubei[17]; **g** Meishan D[14]; **h** Buchanan Lake[13]; and **i** Akkamori 2[14]. These sections represent various depositional environments (e.g., terrestrial, shelf, slope, and pelagic) from multiple locations in the paleo-Southern Hemisphere (Sydney and Karoo basins), paleo-Equator (South China) and Paleo-Northern Hemisphere (Northwest China, Northeast (NE) Pangea, and Panthalassa).

The red and blue dashed lines represent the onset of Hg enrichment and negative carbon isotope excursion, respectively. The gray rectangle represents the extinction interval for each section. The Permian–Triassic transition (P–T transition) in columns c and d represent the transitional interval within which placement of the Permian–Triassic boundary in Karoo Basin is debated (refer to Fig. 4 for more information). The red arrows in panel A represent the distribution of volcanic ash beds yielding U–Pb ages in Bunnerong[35,57,62]. Abbreviations: Sy = system, St = stage, Z = palynozone, M = member; Fa = facies; Changh. = Changhsingian; G. = Griesbachian; P. = Permian; c. = *Playfordiaspora crenulata*, m. = *Protohaploxypinus microcorpus*, p. = *Lunatisporites pellucidus*.

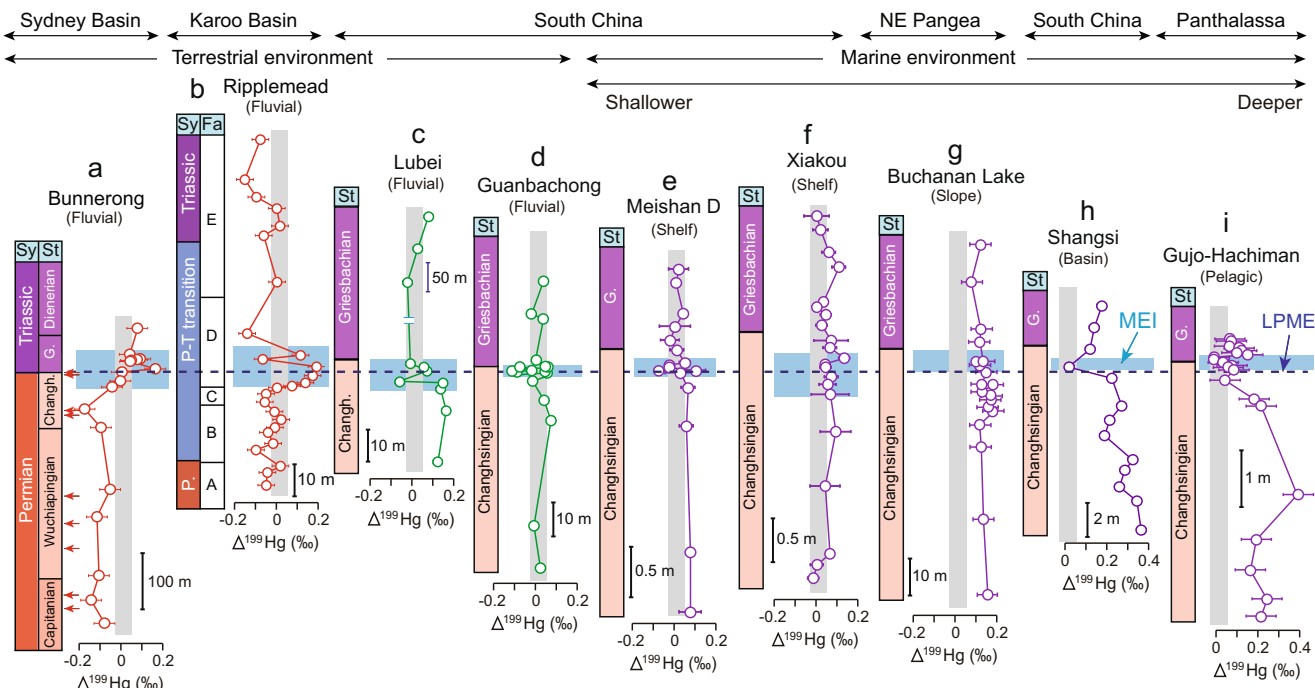

**Fig. 3 | Mass-independent fractionation ($\Delta^{199}Hg$, ‰) for. a** Bunnerong (this study); **b** Ripplemead (this study); **c** Lubei[17], **d** Guanbachong[19], **e** Meishan D[14], **f** Xiakou[14], **g** Buchanan Lake[16], **h** Shangsi[5], and **i** Gujo Hachiman[14]. The red dashed line represents the latest Permian mass extinction (LPME); the vertical gray rectangle represents the $\Delta^{199}Hg$ values of volcanic sources (+0.02 ± 0.06‰[29]). The

horizontal blue rectangle represents the mercury enriched interval (MEI) of each section. The horizontal bars of the $\Delta^{199}Hg$ profiles represent standard deviations (2σ), which are smaller than the symbol size for some samples. Abbreviations: Sy = system, St = stage; Fa = facies; Changh. = Changhsingian; G. = Griesbachian; P. = Permian.

conditions (e.g., euxinia), making TOC normalization unsuitable[12,20]. In the Sydney Basin, Hg exhibits a stronger correlation to TOC ($r = +0.60$, $p < 0.01$, and $r = +0.79$, $p < 0.01$, for Bunnerong and Eveleigh, respectively; Supplementary Fig. 4a) than to TS ($r = +0.29$, $p > 0.05$, and $r = +0.44$, $p < 0.01$; Supplementary Fig. 4b) or Al ($r = +0.05$, $p > 0.05$,

and $r = +0.08$, $p > 0.05$; Supplementary Fig. 4c), suggesting that organic matter is the dominant host for Hg. However, in the Karoo Basin, Hg exhibits no or weak correlations to TOC ($r = +0.02$, $p > 0.05$, and $r = -0.03$, $p > 0.05$, for Ripplemead and Bethulie, respectively; Supplementary Fig. 4d), TS ($r = -0.12$, $p > 0.05$, and $r = -0.07$, $p > 0.05$;

Supplementary Fig. 4e), and Al ($r = −0.16$, $p > 0.05$, and $r = +0.00$, $p > 0.05$; Supplementary Fig. 4f; note: $n$ variably 40–152 above), implying the lack of a dominant Hg host phase and making optimal normalization of Hg concentration data uncertain. We chose to normalize Hg to TOC rather than to TS or Al for the sake of uniformity of data presentation with the Sydney Basin sections (this study) as well as other, earlier-studied sites (Fig. 2). However, caution must be exercised in such cases to avoid small denominator values (e.g., TOC < 0.2%) yielding spurious peaks in normalized Hg concentrations (e.g., Hg/TOC[21]).

Substantial Hg enrichments near the extinction interval support excess Hg inputs into both the Sydney and Karoo basins. Increases of Hg/TOC near the Permian–Triassic boundary regardless of TOC content (which ranges from <1% to > 30%; Fig. 2, Supplementary Fig. 2, Supplementary Note 1) document excess Hg inputs in the Sydney Basin (Fig. 2 and Supplementary Fig. 2). Although the dominant host phase(s) of Hg in the Karoo Basin are uncertain, increases in both raw and normalized (i.e., Hg/TOC, Hg/TS, Hg/Al) values indicate Hg enrichment through the extinction interval in the Ripplemead and Bethulie sections (Fig. 2, Supplementary Fig. 3). The observations of significant increases in both raw (~2–10× and ~2–8× for the Sydney and Karoo basins, respectively) and TOC-normalized Hg values (~2–10× and ~4–20×, respectively) for two separate Southern Hemisphere regions that were thousands of kilometers from the STLIP, as well as of the similarity of their Hg records to those of coeval Northern Hemisphere and Equatorial terrestrial (e.g., Dalongkou, Lubei; Fig. 2e, f) and marine successions (e.g., Meishan D, Buchanan Lake, and Akkamori-2; Fig. 2g–i), lead to the inference that these Hg enrichments reflect a global-scale process.

Although $\Delta^{199}$Hg has been widely used to track Hg sources during the Permian–Triassic transition[14,16,18], interpretation of Hg isotopes in sediments can be complicated. Several factors influence $\Delta^{199}$Hg: (1) the proportions of Hg sourced from the mantle (near-zero MIF[22]) versus organic-rich sediments by sills and/or combustion of vegetation and soil by wildfires (negative MIF[23]); (2) the magnitude of Hg fractionation during atmospheric transport, which ranges from 0.1 to 0.8‰ in the modern atmosphere[24]; and (3) mixing of multiple Hg sources, including terrestrial runoff, atmospheric removal, and seawater loading, in terrestrial and nearshore facies[16,17,25,26]. In Permian–Triassic successions, $\Delta^{199}$Hg shows large variations in different settings, with near-zero values in deep-shelf to slope settings but negative excursions in continental, shallow-shelf, deep-basinal, and pelagic settings (Fig. 3). More negative $\Delta^{199}$Hg values in terrestrial (e.g., Lubei, Guanbachong, Fig. 3c, d) and nearshore settings (e.g., Meishan D, Fig. 3e) have been attributed to a larger proportion of terrestrial Hg[16,17,19]. On the other hand, deep-basinal and pelagic facies (e.g., Shangsi and Gujo-Hachiman, Fig. 3h, i), in which terrestrial Hg inputs are limited, provide a promising reservoir for tracking atmospheric fluxes of Hg[23]. For the pelagic Gujo-Hachiman section, the negative or near-zero $\Delta^{199}$Hg values associated with Hg concentration peaks near the LPME, document elevated fluxes of isotopically light Hg because $\Delta^{199}$Hg becomes more positive during atmospheric transport[24]. Combustion of terrestrial organic-rich sediments (e.g., coal[27]), which can lead to elevated Hg concentrations[28] with a negative MIF in marine deposits[29], was the likely dominant source of Hg at a global scale[14] (Fig. 3) during the second stage of the STLIP eruptions[3].

At the Southern Hemisphere study sites examined here, we infer that the positive $\Delta^{199}$Hg values associated with Hg peaks document long-distance transfer of volcanogenic Hg of STLIP through the atmosphere from the Northern Hemisphere, although exact transport vectors cannot be determined. Mercury in terrestrial facies is mainly derived from clastic inputs and atmospheric precipitation[30], which can be readily distinguished based on negative MIF for terrestrially sourced and positive MIF for atmospherically sourced Hg[15,26,30]. The slope of $\Delta^{199}$Hg versus $\Delta^{201}$Hg covariation is between 1.0 and 1.36 for

Ripplemead (Karoo Basin) and Bunnerong (Sydney Basin) (Supplementary Fig. 5), consistent with photoreduction of aqueous Hg(II) driven by dissolved organic matter[15]. Differences in the pre-LPME Permian background $\Delta^{199}$Hg values between these basins (ca. −0.1‰ and 0‰ for Bunnerong and Ripplemead, respectively; Fig. 3) are consistent with the $\Delta^{199}$Hg ranges of modern[29] and ancient terrestrial facies[17,25,29,30] and, thus, probably due to the basin-specific $\Delta^{199}$Hg values of terrestrial materials, or to variable proportions of terrestrial and atmospheric Hg delivered to each study site. However, the significant $\Delta^{199}$Hg increases (-0.1 to 0.2 ‰) associated with Hg peaks close to the Permian–Triassic boundary in both successions support a dominant atmospheric source of Hg to these sites during the LPME interval. Increasing inputs of terrestrially sourced Hg[17,29,31] played a limited role, at most, in generating these Hg peaks because: (1) Hg contents show no or weak correlations to Al (Supplementary Fig. 4); and (2) the positive $\Delta^{199}$Hg shifts associated with the peaks are inconsistent with terrestrial Hg inputs, which generally have negative $\Delta^{199}$Hg values[29]. Given that the two study basins were separated by ~8,000-10,000 km, the similar increases in $\Delta^{199}$Hg (-0.1 to 0.2‰) in each basin are likely to represent a transregional (i.e., global) flux of Hg from a volcanogenic source (Fig. 1). These positive excursions were likely due to photoreduction of Hg(II) during long-distance transport through the atmosphere[24]. Moreover, the sharp decrease of pyrite sulfur isotope ($\delta^{34}$S) values coinciding with the extinction interval in the Bunnerong core provides evidence for increasing atmospheric sulfate concentrations, which have been previously linked to aerosol production by the STLIP[32]. Similarly, the 1783–1784 eruptions of the Laki volcano in Iceland (~70°N) caused coeval climate perturbations in the Southern Hemisphere[33].

Increased delivery of atmospheric Hg to these far-flung basins is unlikely to have been due to emissions from regional or local volcanic sources. Apart from the STLIP, volcanism linked to regional subduction is believed to have generally intensified during the Permian–Triassic transition interval (*sensu lato*) due to the coalescence of Pangea[4]. However, the most active volcanic arcs were distributed around the Paleo-Tethys Ocean (especially its eastern margin), as evidenced by the abundance of volcanic ash beds as well as Hg-enriched beds in the South China area[4,5,19]. In contrast, arc volcanism in the Sydney and Karoo basins was not particularly intense (compared to South China) as Permian–Triassic boundary ash beds there are relatively rare[34]. Furthermore, nine volcanic ash beds that are present in the Bunnerong core around the Permian–Triassic boundary are not associated with Hg enrichments[35] (Fig. 2a), consistent with the hypothesis that they represent regional volcanic eruptions, and thus played a limited role in the Hg enrichments in these basins.

Integrated analysis of Hg concentration and isotope data provides a new tool for inferring the influence of volcanism in both marine and terrestrial sedimentary records. Although negative carbon isotope excursions (NCIE) have been used previously to infer volcanogenic carbon inputs to Permian–Triassic boundary successions[36] (Fig. 2), many processes can emit carbon and alter sedimentary carbon isotopic signals, complicating their linkage to volcanic events[37]. Mercury, on the other hand, can be transferred long distances through the atmosphere at short timescales (months to years[38]). It is rapidly deposited, resulting in a high-fidelity, correlatable signal of volcanic activity in the stratigraphic record. The near-synchronicity among Hg peaks, the onset of the end-Permian NCIE, and the LPME at Meishan D, where all of these features are associated with Bed 25, as well as comparable relationships in many other marine Permian–Triassic sections, points to a causal link between Hg enrichment and changes in the global carbon cycle and marine ecosystems during the latest Permian[14] (Fig. 2). Owing to expanded stratigraphic resolution, the apparent temporal correspondence among these events varies somewhat in terrestrial facies, however, being nearly synchronous in some sections[18,19] and modestly diachronous in others[17,31,39], e.g., Lubei and

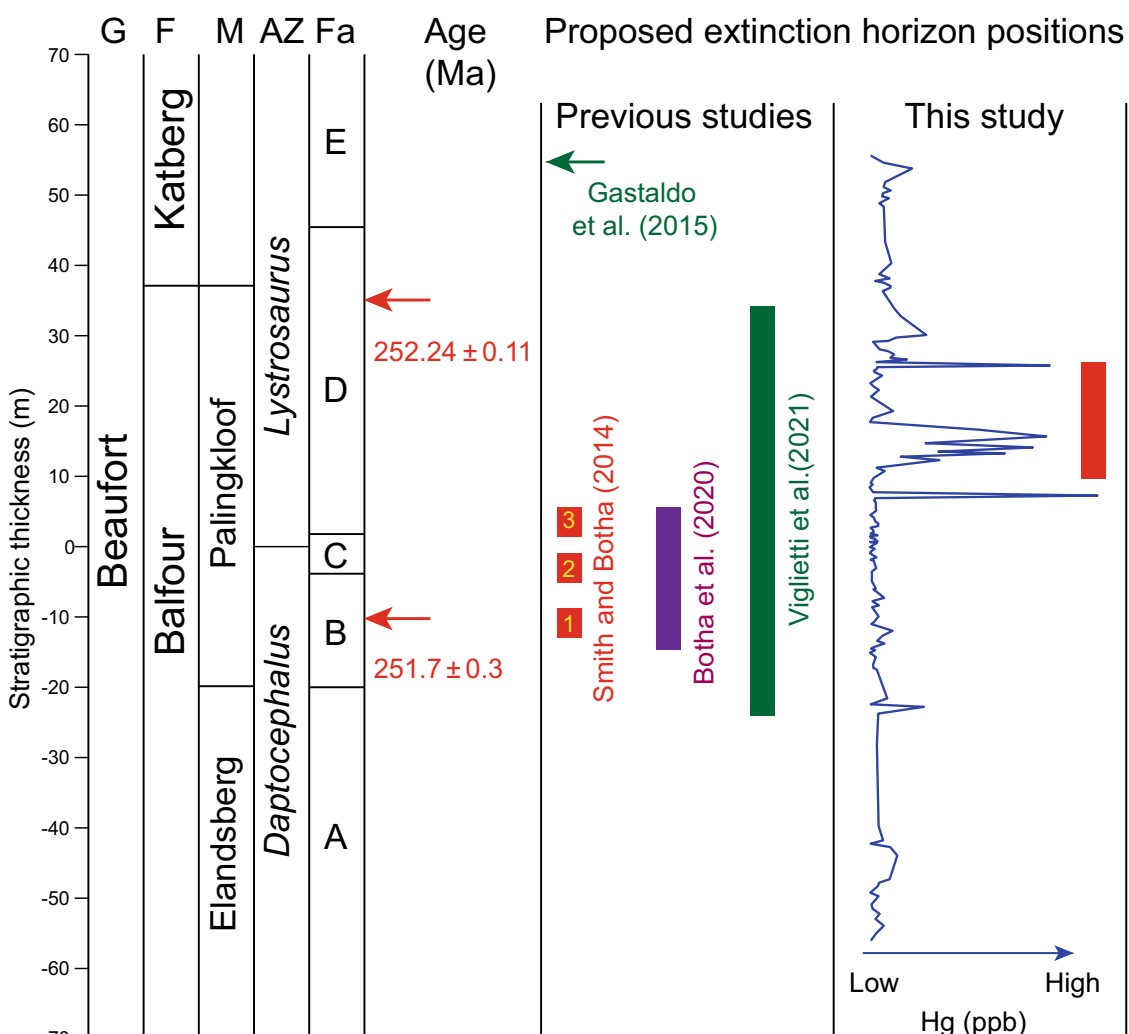

**Fig. 4 | Comparison of previously proposed placements of the tetrapod extinction interval in the Karoo Basin with that of the present study.** The absolute ages of 251.7 ± 0.3 Ma[44] and 252.24 ± 0.11 Ma[45] are radiometric dates from different research groups. Four different levels have been proposed for the LPME horizon, based on either U-Pb data[44,45] or biostratigraphic data[43,46]. The Hg concentration [Hg (ppb)] profile is from the Bethulie section (this study). G = group, F = formation, M = member, AZ = tetrapod assemblage zone, Fa = facies.

Dalongkou, where excess Hg inputs began slightly below the LPME interval and its associated NCIE[17]. Similarly, the initial Hg peaks in the two study cores from the Sydney Basin are also slightly below the terrestrial extinction interval and the onset of the associated NCIE (Fig. 2a, b and Supplementary Fig. 6), representing a temporal lag of ~200 to 300 kyr based on U-Pb dating of the Bunnerong core[35] (Supplementary Fig. 6 and Supplementary Note 2). Volcanogenic Hg inputs to all Permian–Triassic boundary sections appear to have been synchronous at a temporal resolution of ~10⁵ years and, therefore, are likely to have shared a dominant common source (e.g., STLIP). This finding is analogous to that for the Triassic–Jurassic boundary, at which numerous marine and terrestrial sites have yielded Hg anomalies linked to eruptions of the Central Atlantic Magmatic Province at a temporal resolution of ~10⁵ years[23,30,40–42].

The Hg results of the present study provide insights into the stratigraphic positions of the LPME and Permian–Triassic boundary in the Karoo Basin, about which no consensus has been reached to date (Fig. 4). Smith and Botha-Brink[43] proposed a three-phased extinction event in which the major second extinction wave coincided with deposition of a laminated mudstone interval (their Facies C) in the lower Palingkloof Member. Botha et al.[44] generated a U-Pb date of 251.7 ± 0.3 Ma from detrital zircons in Facies B, slightly

beneath the previously proposed Permian–Triassic boundary in the Karoo Basin, and consistent with radiometric dating of marine sections in China and elsewhere (e.g., Australia). However, Gastaldo et al.[45] published a U–Pb date of 252.24 ± 0.11 Ma for a putative tuff bed at the base of the Katberg Formation (Facies E) that resulted in their placement of the LPME some 25 m higher in the succession, within the lower Katberg Formation. Recently, Viglietti et al.[46] undertook a statistical analysis of binned biostratigraphic data that concluded that the LPME had an extended stratigraphic range in the Karoo Basin. Based on widespread evidence from around the world that the initial Hg enrichments in Permian–Triassic boundary sections were associated with the mass extinction event in both terrestrial and marine settings[14,16] (Fig. 1), we propose that the LPME interval in the Karoo Basin is in the uppermost Palinkloof Formation, slightly higher (+10 m, in the lower part of Facies D) than inferred by Botha et al.[44] but substantially lower (−30 to −50 m) relative to that of Gastaldo et al.[45] (Fig. 4). This level represents the upper part of the extinction interval (i.e., third extinction wave) proposed by Smith and Botha-Brink[43], and it coincided with the nadir of the carbon isotope excursion[47] as well as with the rapid faunal turnover phase of the protracted extinction interval proposed by Viglietti et al.[46]. Evidence is mounting that the LPME was not a single bioevent but

multiple extinction episodes over an interval of $10^4$–$10^5$ years in both marine[48] and terrestrial ecosystems[43,49].

Some researchers have argued that the LPME was diachronous between marine and terrestrial successions[35,45], although relatively limited radiometric dating of terrestrial sections, uncertainty regarding the significance of their C-isotope records[50], and the taxonomic dissimilarity of marine and terrestrial fossil records render this inference highly uncertain. Regardless, the onset of Hg enrichments either preceding or synchronous with the mass extinction interval in both marine and terrestrial sections (Fig. 2) supports the STLIP having been a global trigger for this event. The gap between the onset of Hg peaks and extinction interval on land may be due to the stratigraphically expanded nature of terrestrial successions[17,39]. The paleontological record in the Karoo Basin shows the same general pattern: a rich and diverse assemblage of therapsids and reptiles in the Upper Permian based on numerous finds in fluvial deposits[43,51,52]. During the latest Permian, within the *Daptocephalus* Assemblage Zone, a shift from meandering to braided streams in many regions, including South Africa[53], Antarctica[54], and North China[55], was accompanied by the last occurrence of the zonal index fossil *Daptocephalus*, a changeover from non-mammalian synapsids to an archosauromorph-dominated assemblage, and a restructuring of Karoo terrestrial ecosystems. Biodiversity continued to decline until the acme of the extinction near the *Daptocephalus-Lystrosaurus declivis* Assemblage Zone boundary. Regional origination rates increased abruptly above this boundary, co-occurring with high extinction rates to drive rapid turnover and an assemblage of short-lived species symptomatic of ecosystem instability[46], coinciding with the Permian–Triassic transition based on carbon isotope and magnetostratigraphic evidence[47], and concurrent with the peak of environmental disturbance in marine sections as represented by microbial facies in China and elsewhere[56].

Understanding the nature and extent of the environmental footprint of the STLIP is an important issue with regard to better interpretation of the processes surrounding the LPME. In this study, we analyzed Hg concentrations in four terrestrial Permian–Triassic boundary successions in the paleo-Southern Hemisphere as well as Hg isotopes in two of them. Our data show elevated Hg concentrations associated with the biostratigraphically determined extinction interval in all four sections. Positive $\Delta^{199}Hg$ values indicate that the Hg inputs were atmospherically sourced and likely traveled a long distance. Our findings are consistent with the global distribution of Hg derived from the STLIP eruptions. Furthermore, these Hg anomalies allow us to establish a higher-resolution geochronological framework for the extinction interval in the Karoo Basin than has been possible to date from biostratigraphic data alone. Our findings validate the use of Hg data as a volcanic proxy in ancient sediments, demonstrating that peaks in Hg concentrations can serve as a potential chemostratigraphic fingerprint for stratigraphic correlation and event-bed markers in a range of depositional settings.

# Methods

## Study sections

The Sydney and Karoo basins preserve continuous terrestrial successions of Late Permian to Early Triassic age[51,57]. Detailed paleontological work has been carried out in these basins (palynological and vertebrate research in the Sydney and Karoo basins, respectively), allowing detailed studies of the relationships between volcanism, environmental perturbations, and ecosystem effects. The Eveleigh and Bunnerong cores represent high-paleolatitude (~60–90°S) alluvial and coastal plain successions in the Sydney Basin, Australia, and the Bethulie and Ripplemead sections represent mid-paleolatitude (~30–60°S) fluvial successions in the Karoo Basin, South Africa.

The Sydney Basin, located in eastern Australia, contains a >5000-m-thick succession of Upper Carboniferous to Middle Triassic strata[58]. The Upper Permian succession is especially significant in hosting numerous bituminous coal seams that represent some of the world's largest steaming- and coking-coal resources[59]. Coastal exposures of Permian–Triassic transition strata occur in the northern (near Catherine Hill Bay) and southern (near Wollongong) parts of the basin[35,57,60,61]. The Bunnerong (~33.97°S, 151.23°E) and Eveleigh (~33.90°S, 151.19°E) cores are among the most complete Permian–Triassic boundary successions in the Sydney Basin[35,57,62]. A series of seven formations has been recognized, including the uppermost Permian (Changhsingian Stage) Wongawilli Coal and Eckersley Formation, and the lowermost Triassic (Induan Stage) Coal Cliff Sandstone, Wombarra Shale, Scarborough Sandstone, Stanwell Park Claystone, and Bulgo Sandstone. The Upper Permian strata mainly consist of mudstone and coals (deltaic to coastal facies) yielding an abundance of plant fossils (e.g., *Glossopteris*, *Vertebraria*, *Lepidopteris*, and *Dicroidium*). The Lower Triassic strata dominantly consist of sandstones with a few mudstone interbeds (alluvial facies) yielding relatively few fossils. Several volcanic ash beds are preserved in the study sections.

The Karoo Basin is one of most intensely studied areas of Permian–Triassic transition facies. The Ripplemead section (31.49°S, 24.25°E) is located near Nieu Bethesda in the Sarah Baartman District, and the Bethulie section (30.42°S, 26.26°E) is located on the farm Bethel 763 in the Xhariep District of South Africa[43]. These sections, situated in the southwestern part of the main Karoo Basin, contain a ~130 m thickness of the Balfour Formation (including the Elandsberg and Palingkloof members) and the Katberg Formation, which belong to the Beaufort Group (Figs. 2, 4). The Balfour Formation consists mainly of green and gray massive mudstone, greenish-gray siltstone and rare laterally accreted sandstone bodies, whereas the Katberg Formation consists of dark reddish-brown/olive-gray mudstone, siltstone and vertically accreted tabular channel sandstone bodies. The Permian–Triassic boundary extinction interval is marked by a dramatic faunal turnover from the *Daptocephalus* Assemblage Zone[46] to the *Lystrosaurus declivis* Assemblage Zone[43,52]. These sections have been the subject of vertebrate taphonomy, sedimentology, and geochemistry investigations, making them well-suited for the present study due to their stratigraphic continuity and high-resolution biostratigraphic framework[43,47,63].

## Geochemical analysis

Samples were trimmed to remove visible veins and weathered surfaces and pulverized to ~200 mesh in an agate mortar for geochemical analysis. Aliquots of each sample were prepared for various analytical procedures. Organic carbon isotopes ($n = 128$ for Ripplemead) were analyzed at the State Key Laboratory of Geological Processes and Mineral Resources, China University of Geosciences (Wuhan). Samples were reacted offline with 100% $H_3PO_4$ for 24 h at 250 °C, following which the carbon isotope composition of the generated $CO_2$ was measured on a Finnigan MAT 253 mass spectrometer. All isotope data are reported as per mille (‰) variation relative to the Vienna Pee Dee belemnite (VPDB) standard. The analytical precision is better than ±0.1‰ for $\delta^{13}C$ based on duplicate analyses.

Hg concentrations ($n = 70$, 62, 127, and 152 for Bunnerong, Eveleigh, Ripplemead, and Bethulie, respectively) were analyzed using a Direct Mercury Analyzer (DMA80) at the China University of Geosciences (Wuhan). About 150 mg for siltstone samples and 100 mg for mudstone samples were used in this analysis. Results of low-Hg and high-Hg samples were calibrated to the GBW07424 (33 ± 4 ppb Hg) and GBW07403 standards (590 ± 80 ppb Hg), respectively. One replicate sample and one standard were analyzed for every ten unknowns. Data quality was monitored via multiple analyses of GBW07424 and GBW07403, yielding an analytical precision (2σ) of ± 0.5% of the reported Hg concentrations.

Carbon and sulfur concentrations ($n = 128$ and 152 for Ripplemead and Bethulie, respectively) were measured using an Eltra 2000 C–S analyzer at the University of Cincinnati (USA). Data quality was

monitored via multiple analyses of the USGS SDO-1 standard with an analytical precision (2σ) of ± 2.5% of reported values for carbon and ± 5% for sulfur. An aliquot of each sample was digested in 2 N HCl at 50 °C for 12 h to dissolve carbonate minerals, and the residue was analyzed for total organic carbon (TOC), with total inorganic carbon (TIC) obtained by difference.

Major element abundances (*n* = 60 and 99 for Ripplemead and Bethulie, respectively) were determined by X-ray fluorescence (XRF) analysis of pressed powder pellets using a wavelength-dispersive Zetium spectrometer at the State Key Laboratory of Geological Processes and Mineral Resources, China University of Geosciences (Wuhan). Results were calibrated using GSS-35. Analytical precision based on replicate analyses was better than ±2% for major elements and ±5% for trace elements.

A subset of 43 samples (17 and 26 from Bunnerong and Ripplemead, respectively) was analyzed for mercury isotopes using a Neptune Plus multi-collector inductively coupled plasma mass spectrometer (Thermo Electron Corp, Bremen, Germany). The Bunnerong samples were analyzed at Tianjin University[5], and the Ripplemead samples at the State Key Laboratory of Environmental Geochemistry, Institute of Geochemistry, Chinese Academy of Sciences, Guiyang, per methods as reported in Shen et al.[23,30]. Hg isotopic results are expressed as delta (δ) values in units of per mille (‰) variation relative to the bracketed NIST 3133 Hg standard, as follows:

$$\delta^{202}Hg = [(^{202}Hg/^{198}Hg)_{sample}/(^{202}Hg/^{198}Hg)_{standard} - 1] \times 1000$$
(1)

Any Hg-isotopic value that does not follow the theoretical mass-dependent fractionation (MDF) was considered an isotopic anomaly caused by MIF. MIF values were calculated for $^{199}Hg$ and expressed as per mille deviations from the predicted values based on the MDF law:

$$\Delta^{199}Hg = \delta^{199}Hg - 0.252 \times \delta^{202}Hg$$
(2)

Analytical uncertainty was estimated based on replicate analyses of the UM-Almadén secondary standard solution and full procedural analyses of MESS-2.

## Data availability

The authors declare that the main data supporting the findings of this study are available within the Source Data file. Additional data are available from the corresponding author upon request.

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

## Acknowledgements

This research was supported by the Natural Science Foundation of China (92055201, 42072037), National Key R&D Program of China (2022YFF0802900), the State Key Laboratory of Loess and Quaternary Geology (SKLLQGZR2207), and the MOST Special Fund from the State Key Laboratory of Geological Processes and Mineral Resources (MSFGPMR2022-3) to JS, US National Science Foundation (EAR-1636625) to TDF and CRF, National Research Foundation's African Origins Platform and Iziko Museums of South Africa to RMHS, NRF AOP (GUN 136513) and the GENUS DSI-NRF Center of Excellence in Palaeosciences and the Paleontological Scientific Trust (PAST) to J.B. 111 Project from National Bureau of Foreign Experts and the Ministry of Education of China (BP0820004) to J.S. This work is a contribution to IGCP Project 739.

## Author contributions

J.S. conceived the study and designed it; J.S., J.X.Y., R.M.H.S., and J.B. conducted the field work in the Karoo Basin (South Africa); T.D.F. and C.R.F. provided the samples from the Sydney Basin (Australia); J.S. performed the Hg concentration, carbon-sulfur, major element, and

carbon isotope analyzes; J.S. and J.B.C. analyzed the Hg isotopes; J.S., T.J.A., R.M.H.S., and J.B. wrote the paper with significant input from T.D.F., C.R.F., P.D.W., and T.A.M.

## Competing interests

The authors declare no competing interests.
