## [Peer Review File · Nature Communications]

Mercury evidence from southern Pangea terrestrial sections for end-Permian global volcanic effectsReviewer #1 (Remarks to the Author):

Dear authors

Your manuscript on distal volcanic Hg distribution at the End-Permian event and the Permian-Triassic boundary is very interesting as it shows increased Hg-loading in these terrestrial sites. I have some minor comments and suggestions, which you can find in the annotated PDF. Apart from that I only have two issues that I would like to draw your attention to here:

1) Could you please spend a line or two explaining how the Hg is distributed from the c. 70 degrees North position of the Siberian Traps to these two localities situated as far south as 50 - 70 degrees South.

I think it is really good that you discuss other potential, more closely situated, volcanic sources for the Hg at these sites, but your arguments there are not entirely convincing without an explanation of how the Hg would be distributed globally from the Siberian Traps.

2) In your supplement, you have an age-depth model for one of the Australian wells. You state that this is based on radiometric ages and palynological data, but these are not mentioned or discussed. In my experience as a palynologist, terrestrial palynological events are rarely straightforward - hence some explanations would be in order.

All in all a very nice manuscript. I will recommend minor revisions.

Best regards
Sofie Lindström

In their review of the first version of this manuscript, reviewer #1 added some comments to the manuscript file. These comments were forwarded to the authors, who replied as included in this Peer Review File.

Reviewer #2 (Remarks to the Author):

REVIEW OF:

"The global reach of end-Permian volcanism: evidence from sedimentary mercury in southern Pangea terrestrial sections"

by Jun Shen et al.,
submitted to Nature Communications (Manuscript ID: NCOMMS-22-33954-T)

This paper presents mercury spikes in the southern Pangea terrestrial sections on the P/T mass extinction boundary. Moreover, the authors showed that ¹⁹⁹Hg isotope values are characteristic of long-distance atmospheric transfer of volcanogenic Hg, suggesting the Siberian Traps' long-range effects. The manuscript is a fascinating and informative contribution to the field and should be published in Nature Communications. Data is fully presented in the main paper and repository materials and is of good quality. The figures are informative and carefully prepared. However, some minor points need to be improved/added before publication. I recommend publication after minor revision.

Specific comments and minor points:

Authors: There is a lack of the last author in the Supplementary Information file.

Line 48-57. In the opinion of the reviewer, although very important, the issue of dating should be found elsewhere in the article, not at the beginning of the introduction. This issue will not necessarily interest a wide range of Nature Communications readers. In this place, it can be written about the widespread use of Hg as an indicator of volcanism during the most significant extinctions in the history of the Phanerozoic (Hirnantian: Jones et al., 2017. Geology, Smolarek et al., 2019. Sci. Rep., F/F: Racki et al., 2018. Geology, P/T: literature present in the paper, T/J: Percivall et al., 2017. PNAS, K/Pg: Gu et al., 2022. GLOBAL).

In Fig. 1, it appears that the terrestrial and marine location points have been incorrectly marked (red should be terrestrial and blue should be marine).

In Fig. 4, the panel showing high and low Hg concentrations is unclear. It is unknown if it is Hg, Hg / TOC, or Hg / Al. There are no units and, in addition, these are schematic data from one section only.

It would be good to add to the Supp. Inf. a chapter describing the lithology of the examined profiles and the environment of their deposition because it is obvious that they are diverse (e.g., by the content of TOC and TS).

In Fig. S1, the map of China is missing, with the box containing a more detailed map marked.

Point by point response to the comments of the editor and reviewers:

We thank the editor and the two reviewers for comments which have led to significant improvements to our manuscript. We hope we have adequately addressed all the comments from the reviewers and revised the manuscript accordingly. Please note that changes to the text are track-edited (see the WORD file named "main text with marked"), and the line numbers in our response letter refer to those of the revised manuscript.

REVIEWER COMMENTS

Reviewer #1 (Remarks to the Author):

Dear authors

Your manuscript on distal volcanic Hg distribution at the End-Permian event and the Permian-Triassic boundary is very interesting as it shows increased Hg-loading in these terrestrial sites. I have some minor comments and suggestions, which you can find in the annotated PDF. Apart from that I only have two issues that I would like to draw your attention to here:

1) Could you please spend a line or two explaining how the Hg is distributed from the c. 70 degrees North position of the Siberian Traps to these two localities situated as far south as 50 - 70 degrees South.

Reply: It is a really good point. Mercury is transferred through the atmosphere owing to its low vapor pressure (meaning that it is emitted from magmas mainly in gaseous form) and relatively long atmospheric lifetime (0.5-2 years) compared to other metallic species (e.g., Ni, Zn). When the STLIP in northern high-latitude areas released a large amount of Hg, the Hg was distributed at a global scale, showing up in many widely distributed sites (e.g., Hg enrichments near the LPME have been reported from 38 sites, Fig. 1). We have added information about this issue in lines 162-165, 187-811, 205-208.

I think it is really good that you discuss other potential, more closely situated, volcanic sources for the Hg at these sites, but your arguments there are not entirely convincing without an explanation of how the Hg would be distributed globally from the Siberian Traps.

Reply: Thanks, yes, we have considered other potential sources of Hg to the study sites (lines 176-182, 189-200). See our reply above; we have added relevant information in lines 162-165, 187-811, 205-208.

2) In your supplement, you have an age-depth model for one of the Australian wells. You state that this is based on radiometric ages and palynological data, but these are not mentioned or discussed. In my experience as a palynologist, terrestrial palynological

events are rarely straightforward - hence some explanations would be in order.

Reply: Good point. The age model in Figure S6 was dominantly based on radiometric ages (lines 56-63 in SI). The palynological data supplement the geochronological data.

All in all a very nice manuscript. I will recommend minor revisions.

Best regards
Sofie Lindström

Comments from the annotated PDF file.

Line 32, Why not just call it Siberian Traps? And write "Numerous sections on the northern..., proximal to the Siberian Traps...". STLIP is not very reader friendly, I think.

Reply: We prefer use "STLIP" here to emphasize that the Siberian Traps are an LIP. We have defined "STLIP" on lines 30-21, 45, so its meaning should be clear to readers.

Line 55, What about Rampino et al. 2017?

Reply: Modified. We added the reference here.

Line 59, But not only for the PT-transition, also for many other LIPs in general. Could be worth mentioning.

Reply: Modified. We added a mention of it (Hg has been used to track volcanism for other intervals besides the PTB, lines 56-58). We chose to cite review papers here (e.g., Percival et al., 2018; Grasby et al., 2019; Shen et al., 2020) because of the limit on the number of references for NC.

Line 64, It is strange that you do not refer to this paper in the previous sentence too. Afterall, that was where Hg/TOC was first used as a proxy, as far as I know.

Reply: Modified. We added the reference to the previous sentence as suggested.

Line 160, Here you could refer to the Grasby et al 2011 paper on coal fly ash.

Reply: Modified. We added the reference here.

Line 179, In consistent use of LPME and P-T crisis. I suggest that you try to limit the number of variations of names for the crisis.

Reply: Good point. We used LPME for the crisis through the main text.

Lines 187-189, Please rewrite this sentence. I had to read it three times, but I am still not sure I get the meaning correctly. And if I don't there may be others...

Reply: Modified.

Lines 190-191, I think it is really good that you discuss alternatives for the Hg on the southern hemisphere. It would be good to include a line or two about how the Hg would be distributed so far south, when the atmospheric circulation pattern quite effectively would limit major distribution south of the equator. Perhaps the Chenet et al. 2005 paper modelling atmospheric sulfur distribution after the 1783 Laki eruption might help? Laki is situated c. 70 degrees North, similar to the Siberian Traps, but it was a dwarf in comparison. Still their modelling then did indicate some distribution across the equator down to 20 degrees south. Mention height etc

Reply: Many thanks. We added the modern case in lines 187-188.

Lines 227-228, But there, there is also Hg evidence of volcanic activity PRIOR to the NCIEs (e.g. Lindström et al. 2019), as well as U/Pb ages (e.g. Davies et al. 2017). There is actually nothing that says it wasn't the same at the LPME.

Reply: Many thanks. I modified this sentence to make clear that the synchronicity of the LIP and extinction events is at a timescale of $\sim 10^5$ years. Also, we added more references here (lines 222-224).

Lines 239-241, You need to add references here, and if you are referring to this paper, then write that in the sentence.

Reply: Modified. We refer to the records in sections with detailed age-models (e.g., Meishan section) as well as other sections with detailed biozonation schemes. We added references here as suggested.

Lines 244-247, It doesn't really make sense with the 3rd wave of extinctions. I think you need to acknowledge that the LPME was not a fast event that would show up as a horizon in the fossil record, it was an interval, just like the end-Triassic mass extinction. This means that you have to define in this paper, whether you are talking about the onset or the culmination or the top of the extinction interval, and make this clear to the reader.

Reply: Good point. The extinction was not a horizon but an interval (with multiple extinction events at timescales of 10^4 - 10^5 years), both in marine (Yin et al., 2007, GPC) and terrestrial ecosystems (e.g., Smith and Botha-Brink, 2014, PPP; Viglietti et al., 2022 Palaeontology). We made this point clear in lines 81-82, 245-247.

Lines 252-253, But there is no horizon...

Reply: Modified. We changed “extinction horizon” to “extinction interval”.

Line 261, So, does this shift occur at the corresponding level in Australia? Compare with papers by Michaelsen 2002 in *Palaeo3*. You can also find some info about shifts in other Gondwanan localities in Lindström & McLoughlin 2007 in *RevPalbo*

Reply: Modified. We added these references for Antarctica, and rewrote this sentence. However, we did not add the records for Australia for it is still under debate whether a facies shift occurred there. Published data show that, other than a ponding event that formed extensive standing water following the plant community die-off, there was little if any change in fluvial style associated with the P-T transition in eastern Australia (Fielding et al., 2021, *Sedimentology*).

Reviewer #2 (Remarks to the Author):

REVIEW OF:

“The global reach of end-Permian volcanism: evidence from sedimentary mercury in southern Pangea terrestrial sections” by Jun Shen et al., submitted to *Nature Communications* (Manuscript ID: NCOMMS-22-33954-T)

This paper presents mercury spikes in the southern Pangea terrestrial sections on the P/T mass extinction boundary. Moreover, the authors showed that ¹⁹⁹Hg isotope values are characteristic of long-distance atmospheric transfer of volcanogenic Hg, suggesting the Siberian Traps' long-range effects. The manuscript is a fascinating and informative contribution to the field and should be published in *Nature Communications*. Data is fully presented in the main paper and repository materials and is of good quality. The figures are informative and carefully prepared. However, some minor points need to be improved/added before publication. I recommend publication after minor revision.

Reply: Many thanks for the positive comments on our Ms.

Specific comments and minor points:

Authors: There is a lack of the last author in the Supplementary Information file.

Reply: Sorry for the error. We updated the authors in the SI.

Line 48-57. In the opinion of the reviewer, although very important, the issue of dating should be found elsewhere in the article, not at the beginning of the introduction. This issue will not necessarily interest a wide range of *Nature Communications* readers. In this place, it can be written about the widespread use of Hg as an indicator of volcanism during the

most significant extinctions in the history of the Phanerozoic (Hirnantian: Jones et al., 2017. Geology, Smolarek et al., 2019. Sci. Rep., F/F: Racki et al., 2018. Geology, P/T: literature present in the paper, T/J: Percivall et al., 2017. PNAS, K/Pg: Gu et al., 2022. GLOBAL).

Reply: Many thanks for this suggestion. In the first paragraph (lines 45-55), we discuss the background of research on PTB volcanism. In the second paragraph, we discuss Hg as a proxy for volcanic inputs to sediments. We added a sentence to show that Hg has been widely used as a volcanic proxy in lines 56-58. We chose to cite review papers because of the limit on the number of references for NC.

In Fig. 1, it appears that the terrestrial and marine location points have been incorrectly marked (red should be terrestrial and blue should be marine).

Reply: Sorry for the error. We updated the figure.

In Fig. 4, the panel showing high and low Hg concentrations is unclear. It is unknown if it is Hg, Hg / TOC, or Hg / Al. There are no units and, in addition, these are schematic data from one section only.

Reply: Good point. the profile shows Hg concentrations. We updated Figure 4 by addition of the units of Hg, and we added clarifying text to the figure caption in line 594.

It would be good to add to the Supp. Inf. a chapter describing the lithology of the examined profiles and the environment of their deposition because it is obvious that they are diverse (e.g., by the content of TOC and TS).

Reply: Good point. Based on the policy of NC, we present the lithology of the study sites in the "Materials and Methods" section. The text includes a lithologic description of the study sites from the Sydney (lines 293-307) and Karoo (308-321) basins.

In Fig. S1, the map of China is missing, with the box containing a more detailed map marked.

Reply: Many thanks. The map in the Fig. S1 is a geo-map of South China craton during the Permian-Triassic transition. A detailed map of China is not necessary.

-----End-----